# Long-Term Outcomes of Extracorporeal Life Support in Respiratory Failure

**DOI:** 10.3390/jcm12165196

**Published:** 2023-08-09

**Authors:** Filip Burša, Michal Frelich, Peter Sklienka, Ondřej Jor, Jan Máca

**Affiliations:** 1Department of Anaesthesiology, Resuscitation and Intensive Care Medicine, University Hospital Ostrava and Faculty of Medicine, University of Ostrava, 17. Listopadu 1790, 708 00 Ostrava, Czech Republic; filip.bursa@fno.cz (F.B.); michal.frelich@fno.cz (M.F.); peter.sklienka@fno.cz (P.S.); ondrej.jor@fno.cz (O.J.); 2Institute of Physiology and Pathophysiology, Faculty of Medicine, University of Ostrava, Syllabova 19, 703 00 Ostrava, Czech Republic

**Keywords:** extracorporeal membrane oxygenation (ECMO), extracorporeal life support (ECLS), respiratory failure, ARDS, long-term outcomes

## Abstract

Although extracorporeal life support is an expensive method with serious risks of complications, it is nowadays a well-established and generally accepted method of organ support. In patients with severe respiratory failure, when conventional mechanical ventilation cannot ensure adequate blood gas exchange, veno-venous extracorporeal membrane oxygenation (ECMO) is the method of choice. An improvement in oxygenation or normalization of acid–base balance by itself does not necessarily mean an improvement in the outcome but allows us to prevent potential negative effects of mechanical ventilation, which can be considered a crucial part of complex care leading potentially to an improvement in the outcome. The disconnection from ECMO or discharge from the intensive care unit should not be viewed as the main goal, and the long-term outcome of the ECMO-surviving patients should also be considered. Approximately three-quarters of patients survive the veno-venous ECMO, but various (both physical and psychological) health problems may persist. Despite these, a large proportion of these patients are eventually able to return to everyday life with relatively little limitation of respiratory function. In this review, we summarize the available knowledge on long-term mortality and quality of life of ECMO patients with respiratory failure.

## 1. Introduction

Extracorporeal life support (ECLS) is an established method of organ support in critical care patients. In cases of severe acute respiratory failure (ARF) or acute respiratory distress syndrome (ARDS), the use of veno-venous extracorporeal membrane oxygenation (VV ECMO) is usually considered [1]. Although the effectiveness and usefulness of the method have been confirmed, especially after the epidemics of respiratory viral diseases [2,3], the method is still associated with a high risk of complications. ECMO-supported patients with severe ARF still have a high mortality rate [4]. When assessing ECMO mortality, it is necessary to take into account that without the use of VV ECMO, the majority of these patients are at a high risk of death and that without ECMO support, their outcome would likely be worse—after all, the general mortality rate of ARDS patients is around 40% [5]. The costs of ECMO support in patients with a severe course of the disease are high, especially when ECMO support lasts many weeks or even months. Moreover, technical, financial, and/or personnel resources may not always be available. Hence, it is necessary to use the system effectively and identify patients who might benefit from this support. The clinical outcomes of the treatment of ECMO patients are the best parameters for assessing the treatment effectiveness. Ultimately, the goal of ECMO support should, therefore, not just be the survival of the patients but their return to normal life, to their work and families. Good long-term outcomes count among the most important goals of ECMO support. We should evaluate the success of the ECMO program using not only basic parameters such as mortality (i.e., ICU mortality, 30-day mortality) and the length of hospitalization or weaning from ECMO support, but also parameters including long-term survival or neurological, psychological, and social outcomes. This review aims to describe the clinical results of patients who underwent VV ECMO support for severe ARF or ARDS with an emphasis on long-term outcomes.

## 2. Methods

The search of the Medline database was performed on 25 May 2023 using the advanced search option and the following search keywords: (((extracorporeal) AND (veno-venous)) AND ((longterm) or (long-term)) AND (outcome). The search was limited to the last 10 years; only human studies written in English were included. In total, 37 results were identified for retrieval and full evaluation (Figure 1). All search results, including abstracts, were exported into an external file for further analysis (Appendix A). Two researchers independently read all the identified papers. Only papers containing significant information on the topic are referred to in this review.

Particular attention was paid to the long-term (months or years) outcomes of VV ECMO patients after ARF or ARDS. Mortality, fitness, respiratory complaints, neurological complaints, self-sufficiency, quality of life, and the capability to return to work were considered important outcomes.

## 3. Mortality in VV ECMO Patients

Mortality rates are usually the main outcome of ECMO studies. Still, only two randomized control trials in adult ARDS tested the effectiveness of VV ECMO support and mortality rates in patients on ECMO [2,6]. In the CESAR trial, mortality rates (defined as death at ≤6 months or before discharge) for ARDS patients treated with ECMO were 37% compared to 53% in conventionally managed patients (RR 0.69; 0.05–0.97, *p* = 0.03) [2]. In that study, only 76% of patients meeting the criteria for ECMO support in regional hospitals who were subsequently transferred to an ECMO center met the same criteria after an initial assessment in the center, which points to the beneficial effect of treatment in ARDS centers experienced with high volumes rather than to the effect of ECMO support. Another limitation of that study was the lack of a standardized protocol for mechanical ventilation in the control group; moreover, the study did not distinguish between deaths caused by the disease that led to the need for ECMO support and deaths directly related to ECMO cannulation or other ECMO-related complications. The ECMO to Rescue Lung Injury in Severe ARDS (EOLIA) trial compared conventional low VT ventilation to VV ECMO combined with ventilation using an even lower VT (decreased to maintain plateau pressure of 24 cm H_2_O) and lower airway pressures (positive-end expiratory pressure, PEEP, of 10 cm H_2_O and driving pressure of ≤14 cm H_2_O). Rescue ECMO was, however, applied to 28% of the control group patients who had developed refractory hypoxemia. The study was stopped by the safety monitoring board because of futility after 75% of the maximal sample size (reaching the target of 20% absolute risk reduction in mortality was unlikely). Although the decrease in 60-day mortality in the ECMO arm (35% vs. 46%, *p* = 0.09) was not statistically significant [6], a post hoc analysis found that very severe ARDS patients (Pao_2_/Fio_2_ < 50 mm Hg for >3 h; Pao_2_/Fio_2_ < 80 mm Hg for >6 h) had a high probability of benefiting from ECMO in the 60-day mortality [7]. The authors concluded that using ECMO in patients with severe respiratory failure appears to be safer than mechanical ventilation, even if protective lung ventilation is used [7]. A meta-analysis by Munshu et al. pooled mortality data from 429 patients enrolled in CESAR [2] and EOLIA [6] trials and found significantly lower 60-day mortality in the ECMO group compared with the control group (34% vs. 47%; *p* = 0.008) [8]. Another meta-analysis of the results from CESAR [2] and EOLIA [6] focusing on patients with severe ARDS favored the use of ECMO thanks to the improvement in 90-day mortality compared with conventional management (36% vs. 48%; *p* = 5 0.013) [9]. The indication criteria for ECMO in the EOLIA trial were more stringent (a detailed protocol of ventilation optimization prior to indication for ECMO support) than in CESAR, so the same severity of the illness was managed differently. The baseline characteristic of the patients was comparable in both studies. The advantage of this meta-analysis lies in the proper assignment of patients to the given groups (only patients in whom ECMO had been initiated were present in the ECMO arm, and only patients without ECMO support in the control arm). Friedrichson et al. performed a retrospective analysis from a German registry (Federal Statistical Office of Germany) on 45,647 patients in 231 hospitals between 2007 and 2018 [4]. The highest hospital mortality was recorded in 2008: 70.1% among all VV ECMO (*n* = 649) patients and 70.4% among VV ECMO patients with ARDS (*n* = 138). In 2018, hospital mortality decreased in VV ECMO patients with and without ARDS to 53.9% (*n* = 1493) and 54.4% (*n* = 926), respectively. The annual case volume in many of these ECMO centers was low, with a median of four cases for VV ECMO and nine for VA ECMO in 2018, which could have influenced the results due to the lesser experience of the ECMO personnel (note that the median annual number of cases in the centers contributing to the Extracorporeal Life Support Organization (ELSO) registry was much higher, namely 18 cases per center and year [10]). The decrease in mortality over time may be due to the greater experience of the centers and the simultaneous increase in the number of patients with ECMO support. Muguruma et al. also showed a significantly lower mortality rate in patients treated in high-volume ECMO centers (50.4%) than in those treated in low-volume centers (62.5%) [11]. According to the ELSO registry, mortality in adult ARDS patients receiving ECMO from 2015 to 2020 was only 39% [10]. 

At the beginning of the COVID-19 epidemic, extreme mortality rates of VV ECMO COVID-19-associated ARDS (CARDS) patients of up to 94% were reported in some hospitals [12]. On the other hand, data from the ELSO registry from 2020 were more optimistic, reporting an in-hospital 90-day mortality of 37.4% in a large study with 1035 ECMO patients [13]. Karagiannidis et al. reported an in-hospital mortality of 73% in a group of 768 ECMO-supported patients with CARDS from multiple centers in Germany [14]. This result could be, however, associated with the characteristics of their study group (higher age, prolonged median time from the onset of mechanical ventilation to the initiation of ECMO, increased proportion of patients needing dialysis compared to the ELSO group) or heterogeneous experience with ECMO treatment among the personnel of participating ICUs. Moreover, especially during the COVID-19 pandemic (certainly not only in Germany), indications for the initiation of ECMO were influenced by individual attitudes, emotions, and the healthcare system’s strategies (resources and reimbursement), often driven to a large degree by media expectations. It is possible that ill-considered indications for ECMO support, especially in combination with lower experience with ECMO in smaller centers, could have contributed to the high mortality reported in some studies [12,14]. The majority of patients with CARDS were obese, but the 90-day mortalities (48.5% obese vs. 45.5% non-obese, *p* = 0.603) were not significantly influenced by BMI (*p* = 0.47, *p* = 0.771, respectively) [15]. Age < 50 (RR 2.14) and history of chronic immunosuppressant therapy (RR 2.11, *p* = 0.009) were the obesity-associated risk factors for adverse outcomes. Higher doses of corticosteroids (RR 0.57, *p* = 0.05) were associated with a better outcome. Urner et al. demonstrated the association of ECMO support with reduced 60-day mortality in CARDS patients compared to conventional mechanical ventilation in a large multicentric observational study in patients with PaO_2_/FiO_2_ ratio < 80 mmHg [16]. Mortality in the ECMO group was 26% compared to 33% without ECMO (risk difference −7.1%; risk ratio 0.78). ECMO was most effective in patients under 65 years with driving pressure > 15 cmH_2_O or with PaO_2_/FiO_2_ ratio < 80 mmHg. However, some patients were lost to follow-up in that study as they were transferred to a long-term ICU because of the lack of self-sufficiency. The long-term outcome of these patients was, therefore, not known, and their long-term mortality could be higher than stated. Another large observational study was performed in the United States by Shaefi et al. in severe CARDS cases (patients with PaO_2_/FiO_2_ ratio < 100 mmHg), with a combined endpoint of 60-day survival or hospital discharge [17]. Of 5122 critically ill patients, 1297 patients were eligible for the target trial (meeting the inclusion criterion of PaO_2_/FiO_2_ ratio < 100 mmHg in the first 7 days of ICU admission). At 60 days, 35% died in the ECMO group compared to 47% in the non-ECMO group (hazard ratio 0.55; 95% CI 0.41–0.74). 

Enger et al. prospectively followed up on 553 VV ECMO patients from the Regensburg ECMO Centre from 2007 to 2016 [18]. The overall mortality was 52%, with 12% dying after discharge within the follow-up period (mean follow-up of 4.8 years). Survival rates of discharged patients at 30-day, 90-day, 1-year, and 5-year follow-ups were 99%, 95%, 86%, and 76%, respectively. Multivariate analysis showed that higher age, immunocompromised status, higher pre-ECMO SOFA scores (the Sequential Organ Failure Assessment score), and longer duration of ECMO support were associated with higher long-term mortality, whereas out-of-center ECMO cannulation was associated with improved survival, which highlights the importance of avoiding the transport of such patients without ECMO. Another study reported a 6-month mortality of about 40% [19]. 

Provaznik et al. investigated the effect of age on long- and short-term outcomes in a 14-year-long retrospective study including 755 VV ECMO patients. Duration of the ECMO support (median of 8–10 days, *p* = 0.256) and weaning rate (68.2–76.5%; (*p* = 0.354)) were comparable in all age groups, i.e., the higher age was not a risk factor [20]. In-hospital mortality was, however, significantly higher with increasing age (<50 years—30.1%; 50–59.9 years—37.1%; 60–69.9 years—45.6%; ≥70 years—51.8%; *p* < 0.001), increasing age was also associated with worse neurological outcomes. Bacterial pneumonia was generally the most common cause of respiratory failure, although, in the age group of >60 years, aspiration was the most common cause for the need for ECMO. Multi-organ failure was the most common cause of death in the younger age groups, while in the older age groups, secondary respiratory failure (without renewing ECMO support) was the most common cause of death. Interestingly, C-reactive protein (CRP) on Day 1 of ICU hospitalization was inversely associated with mortality, which could have been caused by the exhaustion of a patient’s immune reaction. Risk factors for death during the follow-up also included coronary artery disease, primary lung disease, malignancy, hemodialysis (chronic hemodialysis was a stronger risk factor for in-hospital mortality than acute kidney failure requiring hemodialysis), immunosuppression, and bilirubin levels. Higher bilirubin levels were associated with worse outcomes by Masha et al. [21], too. The association of lower CRP with higher mortality in ECMO patients was also reported in another study [22].

In complex ECMO centers, the ability to transport patients on ECMO support is an inherent part of care. This is because conventional transport of patients with severe respiratory failure on mechanical ventilation could lead to profound hypoxemia, acidosis, or severe complications of aggressive ventilation, which is likely to occur during transfer without ECMO (e.g., pneumothorax, right ventricular failure, etc.). Burrell et al. compared the 6-month mortality in patients who were ECMO-cannulated in referral hospitals with those who were primarily admitted to an ECMO center [23]. There was no overall difference in survival at 6 months between the retrieved and ECMO center patients (75 vs. 70%, *p* = 0.690), which indicates that transport on ECMO is safe.

In all, the 60-day mortality of VV ECMO patients is approximately 35% in most studies, and the mortality in the horizon of several years approx. 50%, with some exceptions reporting both superior and inferior results. The comparison of mortality reported in ARDS patients with and without ECMO support in multiple studies is shown in Table 1.

## 4. Functional and Psychological Impairment in Survivors of VV ECMO and Return to Work

After the initial stabilization of critically ill patients undergoing VV ECMO support, their ability to recover and their overall condition in the horizon of months and years is a critical question. It is, of course, necessary to keep the patients in the best possible condition, including active rehabilitation. Prevention of the failure of other organs and maintaining an adequate nutritional status are challenges in everyday practice. At the same time, a number of technical ECMO complications can occur. Bleeding complications or the necessity to frequently change ECMO circuits are not exceptional.

McDonald et al. investigated the quality of life by interviewing ECMO survivors in whom ECMO was needed due to respiratory failure [24]. Forty-two patients completed the interview, with a median period after weaning from ECMO of 14.6 months. A total of 62% of patients were highly self-sufficient (according to the Katz Index of Independence of Activities of Daily Living). Nevertheless, anxiety was present in 48% and depression in 26% of these patients. A systematic follow-up of survivors to identify patients with problems could help prevent further deterioration of their problems, which are not only psychological, and should ideally constitute a part of the center’s care. 

Wilcox et al. examined in their meta-analysis the long-term morbidity, specifically the health-related quality of life (HRQL) [25]. They reported greater decrements in HRQL in ARDS survivors managed with ECMO compared with survivors of mechanical ventilation. Interestingly, ECMO survivors suffered significantly less from depression and anxiety than those managed solely using mechanical ventilation. HRQL assessment was also investigated in a study by Rilinger, who retrospectively analyzed 289 VV ECMO patients in the Freiburg ECMO Centre in Germany hospitalized between 10/2010 and 06/2019 [26]. Hospital survivors (hospital mortality 55.4%) showed a high 6- and 12-month survival rate (89% and 85%, respectively); moreover, the estimated survival rate of those discharged alive from ICU after 9.7 years was 68.5% (95%-CI 56.9–80.1%). Underlying pulmonary disease, long-term oxygen therapy, mechanical ventilation of more than 7 days before ECMO initiation, and the duration of ECMO support itself were associated with lower 6-month survival. In a logistic regression analysis, the duration of ECMO support was the only independent predictor of 6-month mortality. Overall, high levels of HRQL were reported in survivors (the median total score of the 36-Item Short-Form Health Survey was 73), with limitations predominantly in the questionnaire categories physical, role limitations, physical health, and general health. There were no limitations in the domains of emotional situation and social functioning. No association between HRQL and the duration of ECMO support was detected. Mental limitations were low (Hospital Anxiety and Depression Scale; HAD-D score of 2 and HAD-A score of 3). Overall, 82% of these ECMO survivors returned to work; 61% continued in their previous jobs, 21% had to change their jobs, 8% were permanently disabled, and 10% were already without work before ECMO support.

In their retrospective study covering a period from 2005 to 2019, Gray et al. reported data from 125 VV ECMO patients from a Sydney ECMO center [27]. Survival to discharge was 70%, and in more than half of patients, regular follow-up was completed (median of 11.8 months). Most problems (especially weakness and breathlessness) were resolved within six months, and 60% of patients returned to work within that period. Of patients who were previously employed, 83% had returned to any work within 6 months. Symptoms of depression, anxiety, or post-traumatic stress disorder (PTSD) persisted longest, up to 36 months in 33%; it is, however, necessary to say that 14% of the patients had psychological problems already before ECMO. Another French study compared a 1-year outcome in ECMO vs. non-ECMO survivors of ARDS associated with 2009 influenza A (H1N1) [28]. Surprisingly, mortality in the ECMO group was 36% compared with 28% in the non-ECMO group; however, the severity of ARDS was not mentioned in their study, so it could be speculated that the study was burdened with a selection bias—patients in the ECMO group might have suffered from more serious ARDS. A total of 50% of the ECMO group and 40% of the non-ECMO group reported significant exertion dyspnea, 83% and 64% had returned to work, and 75% and 64% had decreased DL_CO_ (diffusing capacity of the lung for carbon monoxide), despite their near-normal and similar lung function test results. HRQL assessed by the SF 36 did not find any differences between groups. The same can be said of anxiety (50% and 56%) and depression (28% and 28%) as well as a risk for post-traumatic stress disorder (41% and 44%, respectively). More than half of the survivors returned to work; this finding is also consistent with other studies reporting similar findings about anxiety or PTSD [29,30,31].

It could be, therefore, concluded that the majority of patients who were disconnected from VV-ECMO and discharged from the ICU are self-sufficient in the long term, and more than three-quarters of them return to work (Figure 2). 

## 5. Pulmonary Recovery

Pulmonary function was normal to near-normal [32] in a 5-year follow-up study of ARDS ECMO survivors by Herridge et al. Similarly, pulmonary limitations (measured by St. Georges Respiratory Questionnaire; SGRQ) were reported to be very low in the above-discussed study by Rilinger et al. [26] and were comparable to the limitations prevalent in the general population. ECMO survivors showed significantly lower levels of limitation in the SGRQ categories Impacts, Symptoms, and Activity than the COPD (chronic obstructive pulmonary disease) cohort but higher levels of limitation compared to the general population. In the PRESERVE study, 36% of patients after ECMO reported persistent dyspnea at the 6-month follow-up (e.g., shortness of breath even during light exercise), which is comparable to overall results in ARDS patients [33]. In addition, 30% of patients were still on pulmonary medication (long-acting b2-agonists, inhaled corticosteroids, home oxygen therapy). The SGRQ scores indicated that patients with longer ECMO support had more pulmonary symptoms. In the study by Gray et al., the resolution of lung function abnormalities took up to 12 months [27].

Linden et al. performed high-resolution computed tomography (HRCT) of the lungs in 21 ARDS ECMO patients [34]. The majority of these patients (76%) had residual lung parenchymal changes on HRCT, namely reticular patterns in combination with parenchymal distortion, representing together interstitial fibrosis. Ground-glass opacities with architectural distortion similar to that found in fine fibrosis were found in 24% of subjects. According to the pulmonary function and the exercise test, 43% of subjects had signs of a mild obstructive disorder (FEV1 < 80%), a mild restrictive pattern (TLC < 80%) was found in 14% of patients, and a lower normal interval was found in 43% of patients. Of those who failed to complete the test, 76% interrupted the exercise due to leg fatigue and 24% due to dyspnea. DL_CO_ was reduced in 65% of patients. Other studies, however, indicate very good lung repair with minimal long-term changes in CT scans [28,35].

In all, the limitation of pulmonary function diminishes or disappears over several months or within a year after the patients are discharged. Although the limitation to lung function is typically mild after a year, complaints comparable to those in COPD patients may persist in some patients. 

## 6. Long-Term Outcomes in Children

VV ECMO can be used also in pediatric cases of severe respiratory failure. Mallory et al. described the outcomes of 202 VV ECMO patients in highly experienced pediatric centers from 2011 to 2016 [36]. A total of 67% of patients survived until discharge from the ICU; tracheostomy was performed after ECMO decannulation in 14% of survivors, and 9% of survivors were discharged on long-term mechanical ventilation or home ventilation. 

Immunocompromised and non-immunocompromised children up to 18 years old with ARDS were evaluated in a French study with 111 participants [37]. The survival rate at 6 months after the intensive care discharge was significantly lower in immunocompromised patients compared to non-immunocompromised ones (41.7% vs. 62.8%; *p* = 0.04). ARDS severity was similar between the two groups. 

## 7. Prediction of the Outcome

Schmidt et al. created the PRESERVE score to identify factors associated with death within 6 months after ICU discharge and to develop a practical mortality risk score in ARDS patients on VV ECMO. Their study was based on data from 140 patients treated with ECMO from a French ICU in 2008–2012 [33]. A total of 60% of them were still alive after 6 months; the median overall follow-up was 17 months (11–28). The score analyzed pre-ECMO parameters for providing a risk estimate, namely the age, body mass index, immunocompromised status, prone positioning, days of mechanical ventilation, sepsis-related organ failure assessment, plateau pressure, and positive end-expiratory pressure. The PRESERVE classes 0–2, 3–4, 5–6, and ≥7 correspond to cumulative probabilities of 6-month survival of 97, 79, 54, and 16% (*p* < 0.001). HRQL evaluation of the 6-month survivors revealed satisfactory mental health in 80%, but persistent physical and emotional-related difficulties, especially anxiety and depression, were relatively common. Of these 67 patients with satisfactory mental health, 7 were retired, 3 were unemployed, and 57 were working full-time before their critical illness. A total of 72% of the latter had returned to work (52% to their previous work), and most reported normal functions. These patients, however, still had substandard SF-36 physical domain scores, particularly in the emotional component (*p* < 0.001). Patients with longer follow-ups (503 days) had significantly improved scores in the domains of role limitations due to physical health and role limitations due to emotional health. The patients were at an increased risk of anxiety, depression, and post-traumatic stress disease (34%, 25%, and 16%, respectively), which is similar to those reported in other post-ICU studies [33]. 

Scoring systems for ECMO mortality prediction in ARDS patients were compared in a retrospective study in patients with and without influenza A [38]. PRESERVE, RESP, PRESET, ECMOnet, Roch, and APACHE II scores were included to predict patient outcome (mortality). Regarding ECMO support success predictions, the area under the receiver operating characteristic curve (AUC) was 0.62 for PRESERVE, 0.44 for RESP, 0.57 for PRESET, 0.67 for ECMOnet, and 0.62 for Roch, respectively, with score calculations performed according to the original papers. APACHE II score was found to have the best predictive capabilities (highest AUC values) of all these scores, which was true for ARDS patients both with and without influenza. The cut-off value for mortality prediction in the APACHE II score was 32 points. To the best of our knowledge, however, there is currently no score predicting long-term outcomes from the perspective of neurological or physical limitations in patients after VV ECMO. 

## 8. Discussion

Limited information is available about the long-term outcome of patients suffering from acute ARDS or ARF on VV ECMO. Most available studies focused on mortality and short- to mid-term follow-up.

The Extracorporeal Life Support Organisation (ELSO) database provides valuable information about patients on ECMO. However, the data may be biased. It is a paid-membership organization bringing together predominantly high-volume centers. Membership includes entering detailed information on patients into the database, which is, therefore, one of the best data resources in this respect worldwide. ELSO data, statistics, and guidelines thus have a great impact on everyday practice. On the other hand, a majority of minor ECMO centers do not contribute to this database, and, therefore, the success parameters found in the ELSO database records might be biased by the fact that data from ECMO centers with low experience are largely missing in this database. According to the ELSO live registry dashboard (www.elso.org accessed on 19 May 2023), survival to discharge of adult patients with pulmonary ECLS (extracorporeal life support) in all locations (including Europe, North America, Asia, etc.) was 58% (66% survived ECLS, more than 50,000 ECMO runs). When limiting the results to Europe, the survival was 63%. Overall outcomes from 2018 to 2022 in the international summary for adult pulmonary ECLS is 69% (77% survived ECLS). Rather, ELSO data represent the best achievable result when ECMO is performed in experienced centers; however, the results might be burdened with a selection bias—in larger centers with multiple ECMO machines, it is possible that even patients who would not have qualified for ECMO support in a smaller center (i.e., are in better condition) are treated with ECMO. Moreover, in 2018, only 81 centers with 2233 cases were reported in the European ELSO Registry, while in Germany alone, there were 231 centers with 7317 cases reported, which indicates the high level of underreporting in the ELSO registry.

Critically ill patients treated without the use of ECMO also suffer from a number of more or less serious disorders after their discharge from the hospital. Post-intensive care syndrome (PICS) encompasses physical, cognitive, and mental impairments persisting after intensive care unit (ICU) discharge [39]. Pandharipande et al. show in a prospective study that regardless of the reason for ICU admission, the global cognition scores of 40% of ICU patients were 1.5 SD below the population means at 3 months post-discharge and that 26% had scored even 2 SD below the population mean (equivalent to mild Alzheimer’s disease). This impairment persists for more than 12 months and significantly influences patients’ quality of life [40]. Of patients after ECMO support who were transferred to a long-term ICU and after discharge from the long-term ICU, 30% were transferred to a rehabilitation unit, 20% to a skilled nursing facility, 23% to an acute care hospital, and 27% of these patients were discharged home as self-sufficient [41].

The long-term outcome of ECMO patients may be influenced by many factors, some of which may be unrelated to the disease itself, such as the patient’s management in the ECMO center or even in the pre-ECMO period. In ARDS patients, ECMO is typically used primarily as a “bridge to recovery”. The timely recognition of the severity of the disease by the ICU personnel and contacting an ECMO center is one of the key factors of good outcomes. This is particularly important as aggressive ventilation with high mechanical energy may cause further damage to the lungs (so-called ventilator-induced lung injury, VILI) [42,43,44]. Such lung injury may take longer to resolve or cause even lung fibrosis, which could be prevented by the timely initiation of ECMO support. In addition, it might be beneficial to perform ECMO cannulation in the primary hospital rather than in the ECMO center to avoid patient transport on aggressive ventilation and associated risks. In larger ECMO centers with higher numbers of patients (such as larger university hospitals), the patients can benefit from the greater experience of the center personnel, the often also better equipment, or the presence of specialists from other fields, allowing the resolution of potential complications (including demanding surgical procedures). Long-term outcomes are influenced not only by specific management in an ECMO center but also by the management in the referral hospital. An ECMO center should, therefore, also strive to implement the strategy of ECMO indication in the cooperating referral hospitals. The outcome of ECMO patients may be, of course, also influenced by post-ECMO care, for which the patient is often transferred to the original referral hospital. 

Long-lasting pulmonary dysfunction after ARDS without ECMO support is also common in patients after the discharge and affects both the structural and functional properties of the lungs. However, a complete resolution of parenchymal infiltrates may occur within the first 12 months after ARDS. However, persistent abnormalities, mainly reticular patterns and ground-glass opacities, are observed in more than half of the survivors [45]. Patients with pulmonary ARDS showed more severe abnormalities on thin-section computed tomography (CT), including ground-glass opacity or reticular density [44], than patients with extrapulmonary ARDS. There was, however, no difference in the quality of life between these two groups. Orme et al. reported improvement in pulmonary function (but without reaching normal levels) with DL_CO_ of about 70% of the norm at 12 months after ARDS [46]. Herridge et al. reported the 6 min walking distance in the 5-year follow-up of the ARDS patients to be 76% of the predicted distance and the physical functioning component of the SF-36 questionnaire to be 82% of the predicted score matched for age and sex [32]. These findings were also confirmed by Masclans et al. [47]. 

Data on pediatric patients are limited, but ICU-acquired weakness occurs in 1.7–4.7% of survivors in the general ICU population, which is lower than in adults [48]. Cognitive dysfunction is also much rarer, occurring only in 3.4% of pediatric ICU survivors [49]. As data on ARDS ECMO pediatric long-term outcomes are lacking, we could infer from the above that the outcomes of pediatric ECMO patients will also be better than those of adults.

## 9. Conclusions

Although the long-term mortality and morbidity of patients with severe respiratory failure requiring ECMO support is still significant, both survival and post-ECMO quality of life have improved, especially in recent years. The majority of the weaned patients are discharged from the hospital and return not only to their homes but also to work. However, although the standard of care improves, these patients may suffer for many months and years from anxiety, sleep disorders, or depression as the most common long-term neurological disorders after ECMO support. Slight pulmonary limitations also often persist, although they tend to resolve over the course of years.

## 10. Future Directions

VV ECMO is a well-established method with great success and good overall outcomes, preventing death in many of these patients. ECMO is not a treatment but organ support for the most severe illness, which implies that the mortality rate can still be expected to be high. This, however, does not mean that the method should not be further improved. More high-quality data from more or all ECMO providers should help to identify factors that could influence overall long-term outcomes. A global high-quality free register of ECMO patients would be ideal. High-quality, high-volume randomized control trials should also be performed to better identify patients who would benefit most from ECMO support. The indication process is now rather center-dependent, based on local experience. Management in high-volume ECMO centers is overall at a very high level, but the pre- and post-ECMO management is also crucial for good outcomes. National guidance on the distribution of ECMO centers, resource allocation, and coordination of collaboration between the centers and regional hospitals should improve the overall quality of ECMO care in given regions. Only adequate pre-ECMO care with timely identification of patients suitable for ECMO support, safe transport to the ECMO center, care in an ECMO center experienced in high volumes, and high-quality post-ECMO care could improve the all-important long-term outcomes.

## Figures and Tables

**Figure 1 jcm-12-05196-f001:**
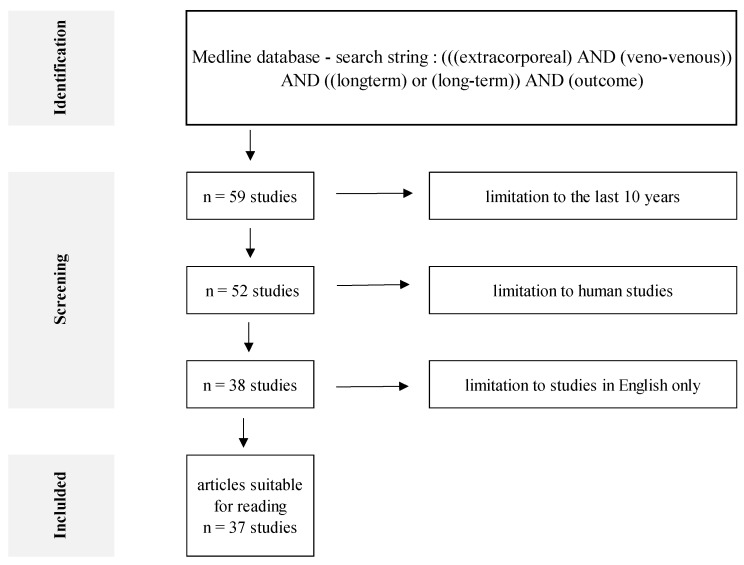
A flowchart of the search strategy.

**Figure 2 jcm-12-05196-f002:**
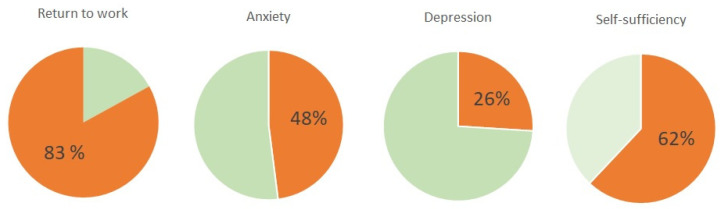
Functional and psychological impairment in survivors reported in [24,26,27,28].

**Table 1 jcm-12-05196-t001:** Mortality of ARDS patients with and without ECMO in selected studies.

Study	Characteristic	Follow-Up	ECMO	Conventional	*p*
CESAR trial [2]	ARDS-H1N1	6 months	37%	53%	0.03
EOLIA trial [6]	ARDS	60 days	35%	46%	0.09
Munshu et al. [8]	ARDS	60 days	34%	47%	0.008
Combes et al. [9]	ARDS	90 days	36%	48%	0.013
Urner et al. [16]	CARDS	60 days	26%	33%	NR *
Shaefi et al. [17]	CARDS	60 days	35%	47%	<0.001

* NR—Not reported; however, as confidence intervals of risk ratios do not include 1, the results were likely statistically significant.

## Data Availability

Not applicable.

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
