# Peer review of "Long-Term Outcomes of Extracorporeal Life Support in Respiratory Failure"

_jcm, 2023, doi:10.3390/jcm12165196_

Round 1

Reviewer 1 Report

Dear Authors,

I appreciate your idea to review the literature available till now about the long term outcome for ECLS (or ECMO) patients.

My main concern is about the type of the manuscript that is not a systematic review, not a meta-analysis and not a narrative review. I suggest to revise the the article and to prepare a systematic review with the appropriate statistical tools for obtaining some evidences.

Minor language polishing and some minor details to be fix (e.g. VV ECMO vs VV-ECMO)

Author Response

Dear Authors,

I appreciate your idea to review the literature available till now about the long term outcome for ECLS (or ECMO) patients.

My main concern is about the type of the manuscript that is not a systematic review, not a meta-analysis and not a narrative review. I suggest to revise the the article and to prepare a systematic review with the appropriate statistical tools for obtaining some evidences.

Thank you for your suggestions and revision of the manuscript. Initially, we were asked to prepare a narrative review article on a given topic for the special issue of Risk Factors and Outcomes of Extracorporeal Membrane Oxygenation in the Journal of Clinical Medicine. According to MDPI guidelines for review, we prepared a narrative review, although, for better clarity, we added the search and publication selection diagram. The manuscript has been amended (changes are highlighted in red in the manuscript), the studies are critically described, highlighting the limitations and strengths of individual studies and their impact on practice. Some tables and figures were added to aid the clarity. It is not possible to rework the paper to a systematic review in the given timeframe. We tried to correct and improve the article to make it a good-quality narrative review. Thank you very much for consider to accept as a narrative review.  

Minor language polishing and some minor details to be fix (e.g. VV ECMO vs VV-ECMO)

The abbreviation was changed to „VV ECMO“ in all text and the text was sent to a professional native speaker proofreader for corrections.

Reviewer 2 Report

Drs. Bursa et al present a review of the available data on long term outcomes following venovenous ECMO for ARDS. Given the recent increase in use of ECMO during the SARS-CoV-2 pandemic this is a welcome and relevant topic. Overall the review covers important data and conveys the information well, but could benefit from improved conciseness and organization. 

- Lines 141-146: The section on COVID-ARDS seems to intrude in the middle of the summary of data on mortality in non-COVID ARDS. Lines 132-141 also cover non-COVID ARDS in the middle of references to COVID ARDS. Consider consolidating the CARDS sections separately 

- there are more studies of survival in CARDS-ECMO than are referenced from later time periods which may provide more complete data (e.g. BMJ

2022 May 4;377:e068723. doi: 10.1136/bmj-2021-068723.)

- Lines 172-183: consider editing for length and removing speculative statements.

- the discussion is confusing and tangential compared to the results section. Lines 319-331 regarding the strengths and weaknesses of the ELSO registers are repetitive from the results section and should either be condensed or limited to one of the two sections. Lines 333-341 should be reported in the results, not in the discussion, if their inclusion is desired. Lines 354-367 are repetitive and should be edited for length.

- Consider addition of a table comparing outcomes in ARDS patients treated with ECMO vs not treated with ECMO for improved comparison. 

No concerns.

Author Response

Drs. Bursa et al present a review of the available data on long term outcomes following venovenous ECMO for ARDS. Given the recent increase in use of ECMO during the SARS-CoV-2 pandemic this is a welcome and relevant topic. Overall the review covers important data and conveys the information well, but could benefit from improved conciseness and organization. 

 Thank you for your suggestions and revision of the manuscript. Some paragraphs were shortened or moved to improve conciseness and organization.

- Lines 141-146: The section on COVID-ARDS seems to intrude in the middle of the summary of data on mortality in non-COVID ARDS. Lines 132-141 also cover non-COVID ARDS in the middle of references to COVID ARDS. Consider consolidating the CARDS sections separately

The CARDS section was reworked and now, it is in one paragraph. 

- there are more studies of survival in CARDS-ECMO than are referenced from later time periods which may provide more complete data (e.g. BMJ 2022 May 4;377:e068723. doi: 10.1136/bmj-2021-068723.)

We added more studies concerning mortality in CARDS, including those you mentioned.

- Lines 172-183: consider editing for length and removing speculative statements.

The sentence was shortened and speculative statements were removed.

- the discussion is confusing and tangential compared to the results section. Lines 319-331 regarding the strengths and weaknesses of the ELSO registers are repetitive from the results section and should either be condensed or limited to one of the two sections.

ELSO register (limitation and strengths) is now commented on only in the Discussion and has been amended.

 Lines 333-341 should be reported in the results, not in the discussion, if their inclusion is desired.

The paragraph was moved to Results.

Lines 354-367 are repetitive and should be edited for length.

The paragraph was reworked and moved. Now, the significance of transport is only discussed in this one paragraph and briefly mentioned in the Conclusions

- Consider addition of a table comparing outcomes in ARDS patients treated with ECMO vs not treated with ECMO for improved comparison. 

The table with the mortality rate in selected studies was added.

Reviewer 3 Report

Thank you for inviting me to review article on hot topic in respiratory medicine:

 Long-term outcomes of the ECLS in respiratory failure. Manuscript is easy to read and all discussions and conclusions presenting huge interest with possible impact on clinical practice.

I have some proposals:

1.     Please add tables and graphics (without graphical part manuscript is difficult to understand)

2.     Please highlight more limitations of all studies

3.     Please add future directions chapter

4.     Also I can suggest but its optional: to change the title. By extending abbreviation ECLS!!!

Author Response

Thank you for inviting me to review article on hot topic in respiratory medicine:

 Long-term outcomes of the ECLS in respiratory failure. Manuscript is easy to read and all discussions and conclusions presenting huge interest with possible impact on clinical practice.

Thank you for your suggestions and revision of the manuscript.

I have some proposals:

  1. Please add tables and graphics (without graphical part manuscript is difficult to understand)

The table (Table 1.) comparing the studies on ECMO mortality was added. In addition, a graphical representation of functional and psychological impairment in survivors was added as Fig. 2.

  1. Please highlight more limitations of all studies

We went through the methods of many studies once again and added the observed limitations and critical comments to the text.

  1. Please add future directions chapter

The Future directions chapter was added.

  1. Also I can suggest but its optional: to change the title. By extending abbreviation ECLS!!!

The abbreviation was replaced by “extracorporeal life support”.

Round 2

Reviewer 1 Report

thanks for revising and re-editing the manuscript according to the comments.

Nothing to add.

Reviewer 3 Report

no other comments